# In Vitro Multiplication and Rooting of Plum Rootstock ‘Saint Julien’ (*Prunus domestica* subsp. *insititia*) under Fluorescent Light and Different LED Spectra

**DOI:** 10.3390/plants12112125

**Published:** 2023-05-27

**Authors:** Lilyana Nacheva, Nataliya Dimitrova, Lyubka Koleva-Valkova, Miroslava Stefanova, Tsveta Ganeva, Marieta Nesheva, Ivan Tarakanov, Andon Vassilev

**Affiliations:** 1Fruit Growing Institute, Agricultural Academy, 12 Ostromila Str., 4004 Plovdiv, Bulgaria; natali.dimitrova@yahoo.com (N.D.); marieta.nesheva@abv.bg (M.N.); 2Department of Plant Physiology, Biochemistry and Genetics, Faculty of Agronomy, Agricultural University, 12 Mendeleev Str., 4000 Plovdiv, Bulgaria; l_koleva2001@yahoo.com (L.K.-V.); andon.vasilev@abv.bg (A.V.); 3Department of Botany, Faculty of Biology, Sofia University, 8 Dragan Tsankov Blvd., 1164 Sofia, Bulgaria; m.stephanova@biofac.uni-sofia.bg (M.S.); tsvetaganeva@uni-sofia.bg (T.G.); 4Department of Plant Physiology, Russian State Agrarian University, Moscow Timiryazev Agricultural Academy, 127434 Moscow, Russia; ivatar@yandex.ru

**Keywords:** micropropagation, shoot culture, photosynthetic pigments, chlorophyll fluorescence, leaf anatomy, stomata

## Abstract

In recent years, light emitting diodes (LEDs), due to their low energy consumption, low heat emission and specific wavelength irradiation, have become an alternative to fluorescent lamps (FLs) in plant tissue culture. The aim of this study was to investigate the effects of various LED light sources on the in vitro growth and rooting of plum rootstock Saint Julien (*Prunus domestica* subsp. *insititia*). The test plantlets were cultivated under a Philips GreenPower LEDs research module illumination system with four spectral regions: white (W), red (R), blue (B) and mixed (W:R:B:far-red = 1:1:1:1). The control plantlets were cultivated under fluorescent lamps (FL) and the photosynthetic photon flux density (PPFD) of all treatments was set at 87 ± 7.5 μmol m^−2^ s^−1^. The effect of light source on the selected physiological, biochemical and growth parameters of plantlets was monitored. Additionally, microscopic observations of leaf anatomy, leaf morphometric parameters and stomata characteristics were carried out. The results showed that the multiplication index (MI) varied from 8.3 (B) to 16.3 (R). The MI of plantlets grown under mixed light (WBR) was 9, lower compared to the control (FL) and white light (W), being 12.7 and 10.7, respectively. In addition, a mixed light (WBR) favored plantlets’ stem growth and biomass accumulation at the multiplication stage. Considering these three indicators, we could conclude that under the mixed light, the microplants were of better quality and therefore mixed light (WBR) was more suitable during the multiplication phase. A reduction in both net photosynthesis rate and stomatal conductance in the leaves of plants grown under B were observed. The quantum yield (Yield = F_V_/F_M_), which represents the potential photochemical activity of PS II, ranged from 0.805 to 0.831 and corresponded to the typical photochemical activity (0.750–0.830) in the leaves of unstressed healthy plants. The red light had a beneficial effect on the rooting of plum plants; the rooting was over 98%, significantly higher than for the control (FL, 68%) and the mixed light (WBR, 19%). In conclusion, the mixed light (WBR) turned out to be the best choice during the multiplication phase and the red LED light was more suitable during the rooting stage.

## 1. Introduction

The European plum (*Prunus domestica* L.) is a very important fruit species in the temperate climate zone with a broad distribution in the southern parts of Europe, but it is also well adapted to the climate in its northern parts. The plum fruits are consumed fresh, dried, canned or are used for brandy making. Plums provide vitamins, minerals and antioxidants, such as flavonoids, carotenoids and glutathione. They are excellent functional foods for cardiovascular health due to their high fiber and potassium content and cholesterol-reducing capacity [1,2]. Regardless of the wide geographic distribution of plums over various climate zones, only specific cultivars could be cultivated in a particular area [3]. The plum cultivation mainly relied on the use of clonal rootstocks because of their various technical and economic advantages [4]. As an example, low vigor rootstocks are indispensable for the orchards with a dense plant system when the planting distance is only 4 × 2 m (1250 trees/ha) [5,6]. The ‘Saint Julien’ (*Prunus domestica* subsp. *insititia*) is one of the most preferred plum culture clonal rootstocks in Europe [6] because it induces low vigor in the tree and is suitable for intensive fruit plum production [3]. Additionally, it develops well on all soil types and a variety of vegetative-covered orchard floors. The other advantage of the ‘Saint Julien’ is that the rootstocks, especially during the first few years of tree development, inhibit the suckers’ growth, and plants start to bear fruits early and provide a regular and good yield with excellent fruit quality [7]. 

The propagation of ‘St. Julien’ by hardwood cuttings is a very profitable technique but the conventional methods of vegetative propagation could retard it. The micropropagation is also an effective alternative method to the mass-multiplication techniques since it provides a large number of rootstocks in a relatively short time [8]. Micropropagation is also an effective large-scale method for in vitro plant multiplication of important insect/disease/virus-free plants which can be propagated in a short time and all year round. It is also a very reliable method for the in vitro preservation of endangered or vulnerable plant species. Micropropagation technology differs significantly from the other agamic propagation methods, since the plants, cultured frequently as microcuttings, can be stored for a long time under constant environmental conditions. The micropropagation of plums under sterile, controlled conditions has been used for more than 40 years to produce a large number of pathogen-free genetically identical plants from selected genotypes [9]. 

Photoperiod, light intensity, light quality, temperature and air relative humidity are factors that are under strict control in the in vitro habitat since they can alter the periodic and oscillator systems upon which a plant’s development depends [10]. The light plays a pivotal role in the plant’s life, not only for photosynthetic energy production but also due to its regulatory role in the molecular, biochemical and morphological processes that govern plant growth and development [11,12,13]. In this context, the artificial light has a crucial role in successful in vitro plant production. Some other factors such as medium composition, gas exchange in the culture vessel and temperature also induce specific physiological responses in the explant. The fluorescence lamps (FL) have been the most used artificial light source in the plant tissue cultures, although the different emission spectra of commercially available lamps do not always match the sensitivity range of plant photoreceptors [14]. In recent years, the light emitting diodes (LED), due to their low energy consumption, low heat emission, specific wavelength irradiation, etc., have become an alternative source of light for plant tissue culture [15,16,17]. 

Numerous studies reported successful applications of LEDs in promoting in vitro growth and morphogenesis in various plant species [14]. Better growth, ex vitro survival rate and biomass yield were reported when various LED treatments were applied [18,19,20,21,22,23,24,25]. In these studies, it was observed that different genotypes have specific requirements towards spectral composition and photosynthetic photon flux density (PPFD). Most of the studies, however, were carried out with herbaceous plants. The available data on the woody species and, in particular, fruit species are very limited. There are some data concerning *Populus* [26] and *Castanea* [27] but the studies on species such as *Pinus* [28,29], coffee [30], *Eucalyptus urophylla* [31], *Cedrela fissilis* [32], *Pinus sylvestris* and *Abeis borisii*-regis [33] have mainly referred to somatic embryogenesis. Very few studies have included woody fruit species [10]. The stimulating effect of the red LED light on the length of the shoots and the leaf area in pear during the in vitro multiplication process has been reported recently [34]. There are only a few reports on the effect of different LED light treatments on plums in in vitro conditions [35]. The aim of the current work was to study the effect of LED light sources (blue, red, white and mixed) on the in vitro growth and rooting of the plum rootstock ‘Saint Julien’ (*Prunus domestica* subsp. *insititia*).

## 2. Results

### 2.1. Multiplication Stage

There were significant differences in the appearance of micropropagated plum plantlets when different light sources were applied (Figure 1, Table 1). 

The plantlets grown under red LED light (R) and those grown under mixed LED light (WBR) had the greatest average stem length, 16.5 mm and 15.1 mm, respectively, although a statistically significant difference with the control (FL) was found only in response to red light treatment (Table 1). The plantlets cultivated in the mixed LED light had the highest fresh (FW) and dry (DW) biomass in comparison to the control plants and to the other light treatments. The shortest stem length and the lowest biomass were observed in plants cultivated under blue light (B). The highest number of leaves was observed under red light (R) but the leaves were almost two times shorter and narrower than those under FL and mixed LED light (Table 1, Figure 1B). The leaves of the plants in the blue light were similar in size to those under the red light but had the lowest counted number.

In the studied plum plantlets, the multiplication index (MI) varied in a wide range—from 8.3 (B) to 16.3 (R)—with the highest value reported in plants grown under red light (Table 1). Under the mixed light (WBR), there was a trend for greater stem length and significantly greater fresh and dry biomass (Table 1, Figure 1).

Photosynthetic pigment content is another important parameter indicating photosynthetic apparatus development, the rate of the photosynthesis and optimal plant growth. It seemed that the blue light (B) had a positive effect on the photosynthetic pigments because the microplants cultivated under blue light showed the highest chlorophyll *a* and total chlorophyll content (Table 2). The higher content of photosynthetic pigments in the leaves of plants cultivated under blue light did not correspond to more intensive photosynthesis (Table 2 and Table 3). The data showed that the content of the chlorophyll *b* did not differ significantly in response to various light treatments. Additionally, the analysis of photosynthetic pigments did not show a statistically significant difference, neither in chlorophyll a/b nor in the chlorophyll/carotenoids ratio. 

In the present experiment, the quality of light had a significant effect on the total phenolic content and antioxidant activity in the plantlets (Table 4). The total phenolic contents values, expressed as gallic acid equivalents (GAE), varied in a wide range from 0.58 to 4.78 mg g^−1^ FW for plants cultivated under FL and WBR, respectively. 

The values of WBR total antioxidant capacities expressed either as % discoloration of DPPH radical reagent or as Trolox equivalent (TE) were twofold higher, 8.59 mg TE/100 mL and 44.09, than the values for FL which were 4.54 and 21.18, respectively. The analyses also showed a correlation between the total phenolics content and the antioxidant activity (*p* < 0.05).

A reduction in both net photosynthesis rate and stomatal conductance in the leaves of plants grown under B were observed (Table 3). The transpiration rate was increased in plantlets raised in B and W light. The highest water use efficiency (WUE, A/E) was observed in the control plants (3.61 µmol CO_2_/mmol H_2_O); in all the other treatments, it was comparable and varied in a range within 65–75% compared to the control.

In the all light regimes studied, the rapid chlorophyll a fluorescence curves had a typical OJIP shape from F_0_ to F_M_ level with distinct J and I phase (Figure 2), indicating that the plum plantlets were photosynthetically active [36].

The minimum (F_0_), maximum (F_M_) and variable (F_V_) fluorescence of the control plantlets and plantlets grown under LED light did not differ significantly (Table 5). The quantum yield (Yield = F_V_/F_M_), which represents the potential photochemical activity of PS II, ranged from 0.805 to 0.831 and corresponded to the typical photochemical activity (0.750–0.830) of the leaves of unstressed healthy plants [37]. This indicated that the applied light sources, even monochromatic light treatments, did not negatively affect the normal function of the photosynthetic apparatus, and, in particular, photosystem II. However, the highest value for the quantum yield (F_V_/F_M_) was reported for the leaves of the plants under mixed light (WBR) and there was a statistically significant difference to the control plant’s yield (FL). 

The probability of the electron transport outside Q_A_ is presented by the ψ_EO_ parameter. The plants subjected to red light treatment had the lowest ψ_EO_ and the parameter values were significantly different from the values in the other treatments. The values of the parameter φ_E0_ followed the same trend.

The performance index (PI_abs_) can be used for the assessment of the PSII state and functional activity in relation to the amount of absorbed energy [38]. In the current study, the highest PI_abs_ was observed in plants cultivated under mixed and blue LEDs, followed by plants from FL and W light treatments. The lowest PI_abs_ was estimated for plants under red light. 

The highest value of total performance index (PI_total_) was marked in the plantlets cultivated under mixed light (WBR) and the lowest was marked in the plantlets cultivated under red light (Table 1). 

The light microscope observation of the transverse sections of in vitro grown *Prunus* leaves revealed a uniform histological organization regardless of the light treatment. The leaf mesophyll was structured in a bifacial manner: the palisade parenchyma was composed of one row of short but easily distinguishable cells, and the spongy parenchyma consisted of three to four layers of more or less tightly arranged cells. 

The epidermis was uniseriate; the height of the adaxial ordinary epidermal cells was greater than the abaxial cells. The stomata, which guard cells was slightly protruded above the epidermal level, were observed only in the abaxial epidermis (Figure 3). 

However, there were some differences in the morphometric parameters of the plants cultivated under the chosen light sources. The leaves from the plants cultivated under broad spectrum light (W and WBR) had the thickest leaf lamina. Similar leaf lamina thickness was observed for the leaves under mixed blue and red light (WBR). Between these two treatments, there was no significant difference but their morphological parameter values were much higher than those of the other three variants. The thinnest lamina was observed in the leaves formed under monochromatic red light (R). A similar result was observed for the mesophyll, since the thickest mesophyll was seen in leaves under white light (W), followed by the leaves under mixed light (WBR). The monochromatic blue light (B) led to the thinnest mesophyll. The highest values for the palisade and the spongy parenchyma were measured in leaves differentiated under white light (W), followed by the leaves of plantlets grown in mixed light (WBR). The only significant difference was estimated for the palisade parenchyma between W and WBR light treatments. The lowest values of the palisade parenchyma were found in the leaves under blue light (B) but they were similar to the other light treatments, red light (R) and fluorescent light (FL), and furthermore no significant difference was noted. The mixed light (WBR) increased the thickness of both adaxial and abaxial epidermises, although the difference for the adaxial epidermis was statistically significant only in FL and R. Red light (R) has the opposite effect, as the adaxial and abaxial epidermises were thinner (Table 6).

In the examined leaves from all light treatments, the stomata had elliptical or round shapes (Figure 4). The largest size of stomata was observed in the leaves under WBR light and there were significant differences in comparison to all the other treatments except for those with W light. The highest stomatal frequency was counted in the control leaves (FL), despite the difference being non-significant between WBR and W variants. The leaves under the R light had the lowest and significantly different stomatal frequency in comparison to all other light treatments (Table 7).

### 2.2. Effect of the LED Lighting on the Rooting

Light had a significant effect on the rooting of in vitro cultivated plantlets of ‘St. Julien’ plum rootstock (Table 8, Figure 5). Under white (W) and blue (B) light, similarly to the control (FL), the percentage of rooting was in a narrow range, from 62.7% to 68.42%. The red monochromatic light (R) stimulated not only the highest percentage of rooting (98.67%) but also the roots were the longest (32.85 mm) under this light. The red light also was the only light regime that provided the development of lateral roots, which reached up to 10 mm. The mixed LED light (WBR) had a suppressive effect, since the plant rooting reached only 19% and the average number of roots per plant was the lowest in comparison to other light regimes (1.40).

## 3. Discussion

Numerous studies have reported the applications of LEDs and made a comparison with the white FL regarding the possible effects on promoting the in vitro organogenesis, growth and morphogenesis of various plant species such as *Gossypium hirsutum*, *Anthurium andreanum*, *Brassica napus*, *Musa acuminata*, etc. [10,22,39,40]. 

The possibility to modulate the light spectrum in accordance with plant demands appears to be one of the most important advantages of the LED techniques [41]. It is known that the photoreceptors, which are responsible for the plant’s development and photosynthesis, have the highest sensitivity and thus are stimulated primarily by the red (R) and blue (B) regions of the light spectrum. In relation to that, most of the studies, with aims similar to the current one, evaluated the impact of the monochromatic R (660 nm), B (460 nm) and combined B (440–480 nm) and R (630–665 nm) lights. According to Gupta and Jatothu [14], recent advances in LEDs technology, in terms of plant growth optimization, are the results of experiments with mixed LEDs rather than only with monochromatic blue or red LEDs.

The plantlets from plum rootstock ‘St Julien’, used in the current study, were cultivated under various light sources. The multiplication index (MI) is one of the most important indicators in plant micropropagation, but it should be considered in a complex manner with other indicators. Under mixed LED light (WBR), the plantlets’ leaf number and size, photosynthetic pigment content and net photosynthetic rate (Table 1, Table 2 and Table 3) did not differ significantly from the control (FL). However, under the mixed light (WBR), there was a trend for greater stem length and significantly greater fresh and dry biomass (Table 1) as well as the highest quantum yield (F_v_/F_M_), although the difference to B was not statistically significant. Considering these indicators, we could conclude that under the mixed light, the microplants were of better quality. Furthermore, the size of the plum explants allows for easier handling and manipulation, which can speed up the process of micropropagation and has a significant practical value. Therefore, we can conclude that mixed light (WBR) is more suitable during the multiplication phase.

Our previous research with raspberry (*Rubus idaeus* L.) [42], highbush blueberry (*Vaccinium corymbosum* L.) [43] and *Pyrus communis* L. [44] showed that a combination of blue, red, far-red and white light (1:1:1:1) stimulated plant growth and biomass accumulation. Similar results were reported by Poncetta et al. [45] who observed that the mixed LED light was less efficient than the fluorescent light in the multiplication of red raspberry, but provided shoots with higher quality. The efficiency of combined LEDs in plant growth and development, when compared to the effect of the monochromatic light, was reported for several other species such as *Lilium* sp. [46], *Chrysanthemum* sp. [47], *Doritaenopsis* sp. [21] and *Lycium barbarum* L. [48]. 

Other researchers have shown that the combination of red and blue LEDs had enhanced the growth of plants from the genera *Mentha* and *Fragaria* [14,19,49]. Muneer et al. [50] also noticed that the combination red and blue LEDs significantly diminished damages caused by the hyperhydricity, especially in carnation genotypes aggravated under fluorescent light.

Along with the intensity of photosynthesis and photosynthetic pigment content, the chlorophyll *a* fluorescence is another indicator of the functional activity of the photosynthetic apparatus of plants. The chlorophyll *a* fluorescence induction has been thoroughly studied since 1931 when Kautsky and Hirsch discovered the negative correlation between the fluorescence intensity and carbon dioxide fixation [51]. The light energy absorbed by plants can have a different fate: to be absorbed by photosynthetic pigments, to be lost as heat due to internal conversion, and to be emitted as fluorescence [52]. The analysis of the induction curves of rapid chlorophyll fluorescence (OJIP test) links the structure and functionality of the photosynthetic apparatus and allows a rapid assessment of plant viability, especially in stress conditions [53]. Previous studies have shown that the parameters of chlorophyll *a* fluorescence in the leaves of plants cultured in a controlled environment could be affected by the light [54,55,56], plant nutritional status [57,58] or environmental stresses [59,60]. The total performance index (PI_total_) presents not only the functional activity of the PSII, but also the PSI along with the rate of electron transport chain between them [38]. The PI_total_ is closely related to the overall growth rate and survival of plants under stress conditions and has been described as a very sensitive and reliable parameter in the JIP test. The highest value of PI _total_ was recorded in the plantlets cultivated under mixed light (WBR) and this accurately corresponded to the highest value of biomass accumulation in plantlets (Table 1).

The increased values of ψ_E0_, φ_E0_, PI_abs_ and PI_total_ parameters that were observed in the plants cultivated under blue light seemed in opposition to their suppressed growth, low net photosynthetic rate (A) and stomatal conductance (gs). One possible explanation could be related to the fact that the measurements were performed on physiologically older leaves than the first fully developed leaves in the other variants in which the fluorescence was evaluated.

Plum is a plant genetically predisposed towards accumulating secondary metabolites of the phenolic group. The phenolic compounds are well known for their antioxidant properties and their synthesis can be stimulated by various environmental factors [61]. The in vitro cultivation, even under carefully controlled environmental conditions, is able to induce, to some extent, oxidative stress in microplants. In such cases, an increased synthesis of various protective molecules, in particular phenolic compounds, can be observed.

It is known that light that affects plant morphogenesis and metabolism is one of the factors responsible for the production of reactive free radicals, such as superoxide anion (O_2_^•−^), hydroxyl radical (OH•) and peroxy radical (ROO•). The free radicals can cause protein denaturation, lipid peroxidation and oxidative DNA damages and negatively affect membrane fluidity [62]. Antioxidants that can scavenge the reactive free radicals can prevent the oxidation of the other molecules and therefore have a protective effect on the cell. In the present study, the mixed LEDs showed the strongest stimulation in the synthesis of phenolic compounds and increased antiradical activity was estimated, respectively (Table 4). Blue and red light spectral regions also have a stimulating effect on the phenolic biosynthesis with a further cumulative effect when they are mixed. Sebastian and Prasad [62] have similar observations about the beneficial effects of red and blue light on plants with induced oxidative stress. In their study, the authors treated rice plants with red and blue light and found that a consecutive application of blue and red light significantly increased the content of phenolic compounds in plants when compared to the control ones that had been cultivated under fluorescent light only.

The studies on gerbera showed that the combination of red (70%) and blue (30%) light with specific light intensity (40–120 μmol m^−2^ s^−1^) could be effective either for modifying the potential of *Gerbera jamesonii* Bolus shoot multiplication, or for controlling the plant morphometry and photosynthetic pigment content [63,64]. Similarly, the proliferation rate of *Brassica napus* in in vitro cultures was higher under blue light than under white light [40]. 

In the present study, the red LED light exerted beneficial effects on the stem length of plum microshoots, number of leaves and multiplication index, although these plantlets had lower fresh and dry biomass in comparison to the control (FL) and mixed light (WBR) plants. These results are similar to the previous observations, which showed that the red light stimulated raspberry shoot elongation at the multiplication stage [42]. In addition, the red light stimulating effect on stem elongation was reported in other species: chrysanthemum [47], grapes [65], *Oncidium* [66], blueberry [67], *Scrophularia takesimensis* [68], stevia [24] and *Carpesium triste* Maxim [69].

According to Manivannan et al. [70], the stimulating effect of red light could be related to the formation of endogenous gibberellins, which are key growth regulators involved in plant cell elongation. Li et al. [71], who studied the effect of red light on grape stem and root elongation, made a similar assumption. The authors suggested that the red light may promote stem growth by regulating the biosynthesis of gibberellins or inducing the expression of an auxin inhibitor gene [71].

As noted earlier, some authors agreed on the positive role of red light, and high R:FR light ratio on shoot proliferation [72]. In addition, R light significantly enhanced the adventitious bud formation and development of *Spathiphyllum cannifolium* [73] and *Mirtus communis* [64]. On the contrary, under monochromatic R or B light in comparison to W or mixed R with B light, Bello-Bello et al. [74] observed a decrease in the proliferation ratio of *Vanilla planifolia*. The same decrease was found by Martínez-Estrada et al. [75] who studied *Anthurium andreanum* and Lotfi et al. [34] who studied *Pyrus communis*. 

The main effects of the R light are explained by its effect on the phytochrome and the synthesis of cytokinins in the plant tissues [76]. The cytokinin biosynthesis opposes the effect of auxins and thus stimulates the development of lateral shoots. The red spectrum also regulates the synthesis of carotenoids and, in particular, strigolactones that seem to affect apical cell dominance by some modification of the auxin fluxes [10,76]. Additionally, the stimulation by the R light seems to be more efficient at the beginning of the multiplication phase. However, different reports assumed that R light alone is not sufficient to activate the chlorophyll synthesis and as a result cause excessive stem elongation and leaf disorders of the so-called “red light syndrome” [77]. The plum plantlets in the current study, which have been cultivated under the monochromatic red light, had the typical “red light syndrome” appearance—long and thin stems accompanied by many, small-sized leaves (Figure 1, Table 1). The red light also had an effect on the anatomy of the leaves; the thinnest leaf lamina, adaxial and abaxial epidermis were measured (Table 7, Figure 3). Similar symptoms of the typical “red light syndrome”, elongated stems, very small leaves and leaves with chlorosis were also observed during the micropropagation of pear plantlets under the red LED light [44]. Unlike the plum plantlets in the present study, the experiment with the pear revealed that the plant raised under red light had the lowest number of leaves as compared to the control (FL), monochromatic blue, white or mixed LED-light-treated plants.

Predictably, the thickest mesophyll was measured in plum leaves formed under white light (W), followed by the leaves of mixed light (WBR), while the leaves that developed under monochrome light were significantly thinner. That observation confirmed the fact that the full light spectrum was beneficial during leaf ontogeny. Additionally, in pepper plantlets and cherry tomato plants, the leaf thickness, and the palisade parenchyma thickness, respectively, were high in leaves developed under RB light [78,79]. The smallest and thinnest leaves were observed in plum plants grown under R light. It could be explained as a reaction to radiation stress during plant development [80]. However, in our study, the mesophyll in all examined variants had the same organization—one palisade layer and three to four layers of spongy parenchyma. The single palisade layer occupied only about a third of the photosynthetic tissue and that ratio remained the same under the different light treatments. In this study, the morphogenesis of the leaf epidermis was not affected by the different light regimes. In all leaves, the ordinary cells and the stomata were well developed. However, the size of the epidermal cells and stomata increased under mixed WBR light. In the variants where the light spectrum included B light, the stomatal frequencies were higher than under monochrome R light treatment. Chrysanthemum leaves grown under mixed RB light had the largest stomata but their number was the smallest out of all other treatments [47]. Controversially, the RB light regime triggered significantly higher stomatal frequency in *Amelanchier alnifolia* leaves compared with those developed under FL light [81]. Studying the effects of R- and B-LEDs on the growth and morphogenesis of grapes, Poudel et al. [65] found that B light was responsible for a higher number of stomata in all the genotypes but that there was no significant difference in the size of stomata under the different light conditions that were tested in the experiment. For both birch and hybrid aspen plants, the R:FR ratio of experimental light treatments did not affect the stomatal density but for silver birch clones grown under extended light spectrum (RGBYO) it was increased [82]. It is presumed that the increased number of stomata on the leaf surface promotes CO_2_ absorption [83] and might facilitate further development ex vitro [84].

According to a number of authors, the blue light is necessary for a proper stomatal opening, can improve the access to CO_2_, and can affect the transpiration and nutrient uptake [85,86,87,88]. 

The blue and red spectra are required for chlorophyll synthesis and foliar growth and their combination in a suitable proportion is important for overall plant growth and development [14]. Increased values of FW and DW of the shoot in ‘St Julien’ plantlets under mixed LEDs (WBR), in comparison to the FL, further implied the necessity of light combination in order to achieve a fine-tuned light spectrum for optimal plant development.

Rooting is an important step in whole plant formation during the micropropagation process and along with the shoot induction has often been evaluated. As aforementioned, the light had a significant effect on the rooting of in vitro cultivated plantlets from the ‘St. Julien’ plum rootstock (Table 8, Figure 5). Under the red monochromatic light (R), the rooting reached the highest percentage (98.67%) with the highest value of root length (32.85 mm). Conversely, under mixed LED light, a very low percentage of plum plants rooted (about 19%). 

Different reports have indicated that the R light alone is effective in root induction, but the effects of different light qualities on root development are often contradictory in the available literature. Kurilčik et al. [89] reported that the monochromatic red light and the red light in combination with fluorescent light improved root development in *Chrysanthemum morifolium* cv. ‘Ellen’. The red light also stimulated the growth of adventitious roots in *Morinda citrifolia* [90] and, according to Ghimire et al. [91], R light improved the root development in *Panax ginseng*. The data of Shulgina et al. [24] showed that the mixed red and blue LED light inhibited the growth of *Stevia rebaudiana* Bertoni shoots, but stimulated root system development.

## 4. Materials and Methods

### 4.1. Plant Material and Experimental Conditions

The experiment was carried out with plum rootstock ‘Saint Julien’ (*Prunus domestica* subsp. *insititia*). 

#### 4.1.1. Establishment of Shoot Multiplication Cultures

Shoots of *P. domestica* subsp. *insititia* ‘Saint Julien’ were collected from the seven-year-old trees growing in the research orchard of the Fruit Growing Institute, Plovdiv, Bulgaria. The explants (apical and stem segments) were taken in mid-April 2019 from young shoots (8–10 cm); their leaves were removed and surface sterilized by immersion in 70% ethanol for 30 s and in a 5% solution of calcium hypochlorite containing 2–3 drops of Tween for 10 min. The shoots were then rinsed four times (10 min each) in sterile distilled water. Nodal segments (10 mm) were cut from the shoots and inoculated on the solid MS medium [92], supplemented with 5 μM BAP, 0.01 μM indol-3-butyric acid (IBA), 30 g L^−1^ sucrose and 6.5 g L^−1^ Phyto agar (Duchefa, Haarlem, The Netherlands). The medium was adjusted to pH 5.6 before autoclaving at 121 °C for 20 min. The in vitro shoot culture was maintained on the above-mentioned nutrient medium at 4-week subculture intervals. Plantlets were placed in baby food glass jars (diameter 60 mm, height 100mm, volume 190 mL) with transparent Magenta B-Cap lids with 25 mL nutrient medium per vessel. In each vessel, five explants were incubated. The cultures were cultivated at 22 ± 2 °C under a 16 h photoperiod (87 ± 7.5 μmol m^−2^ s^−1^ photosynthetic photon flux density, PPFD), provided using a Philips GreenPower LED research module (Philips Lighting,). Four groups of LEDs emitting in white (W), red (R, 650–670 nm), blue (B, 455–485 nm) and far red (FR, 725–750 nm) were used. Plantlets grown in the same conditions but under fluorescent lamps (FL) served as the control. 

Each multiplication experiment used six jars (with five explants in it) for each light treatment and the experiment was repeated twice. 

#### 4.1.2. Rooting Experiment

Apical shoots (10–15 mm long) with two to three leaves from the previous experiment with shoot multiplication were transferred to the rooting medium containing half-strength MS macronutrients, full-strength micronutrients and vitamins, 1.5 μM IBA, 20 g L^−1^ sucrose and 6.5 g L^−1^ Phytoagar, pH 5.6. Plantlets were grown in baby food glass jars at the corresponding light regimes and temperatures described above. Data on the rooting percentage, number of roots per rooted microcutting and the length of roots were recorded three weeks after the beginning of the experiment. Rooting experiments were performed on 30 shoots per light treatment (six jars × five explants) and were repeated twice. 

### 4.2. Physiological and Biochemical Parameters

#### 4.2.1. Growth Parameters

The plants were cultivated for four weeks until their transfer to the fresh culture medium. The analyses were performed after five consecutive passages in total and the following parameters estimating the plants development were measured: fresh (FW) and dry biomass weight (DW), length of the shoots, content of the photosynthetic pigments, total polyphenols, antiradical activity, gas exchange rate and chlorophyll a fluorescence.

The base of the plants was washed with running water to remove traces of the nutrient medium, then dried with filter paper and the fresh weight (FW) of the plants was measured. The dry mass (DW) of the plants was determined after drying at 105 °C (±5 °C) to a constant mass.

A morphological observation of leaf anatomy was conducted. Multiplication index (MI) was calculated as the number of proliferated shoots from one explant. 

#### 4.2.2. Gas Exchange Analysis

A gas exchange analysis was performed with a portable gas exchange system LCpro+ (ADC, Hoddesdon, Herts, UK) at a light intensity of about 130 µmol m^−2^ s^−1^ PPFD and an ambient CO_2_ concentration of 1100 vpm. Each plantlet was placed in the conifer measurement chamber, and the base of the shoot was wrapped in wet filter paper to prevent drying. The net photosynthesis rate (A, µmol CO_2_ plantlets^−1^ s^−1^), transpiration intensity (E, mmol H_2_O plantlets^−1^ s^−1^), photosynthetic water use efficiency (A/E, µmol CO_2_ mmol^−1^ H_2_O) and stomatal conductance (gs, plantlet^−1^ s^−1^) were determined.

#### 4.2.3. Chlorophyll a Fluorescence

Chlorophyll *a* fluorescence induction curves (OJIP) were recorded on each fully developed youngest leaf of five representative plants with a Plant Efficiency Analyser (Handy-PEA, Hansatech Instruments Ltd., King’s Lynn, UK). The analyzed leaves’ spot areas were adapted to dark with special clips for 40 min. The induction curves of the rapid chlorophyll *a* fluorescence (OJIP test) were recorded after illumination with 3000 µmol m^−2^ s^−1^ PPFD for 1 s. The primary data were processed with the PEA Plus Software (V1.10, Hansatech Instruments Ltd., UK). The parameters of the OJIP test (Table 9) were interpreted and normalized according to Strasser and Strasser [93] and Goltsev [94].

#### 4.2.4. Photosynthetic Pigments

Photosynthetic pigments were extracted from fully developed leaves using 0.1 g of fresh plant material in 10 mL of 80% aqueous acetone (*n* = 12). After the filtration of the extract, 1 mL was mixed with 2 mL acetone, and chlorophyll-a (Chl *a*), chlorophyll-b (Chl *b*) and carotenoid (Car) contents (µg g^−1^ FW) were spectrophotometrically determined at absorbances of 663 nm, 645 nm and 470 nm, respectively, according to [95].

#### 4.2.5. Total Polyphenols Determination

The total amount of polyphenol compounds in the plant extracts was determined with the Folin–Ciocalteu reagent [96], according to Singleton and Rossi [97], with slight modifications. The samples (1 g of fresh plant material) were ground with quartz sand and 10 mL 60% acidic methanol, and submerged in an ultrasound bath for 15 min. The homogenized material was left in the dark for 15 h at room temperature for extraction. Afterwards, the test tubes were centrifuged and the supernatant was used for the measurement of total polyphenol content and antioxidant activity. Determination of the total phenolics 40 µL was performed in a mixture of extract, 3160 µL distilled water, 200 µL Folin–Ciocalteu reagent and, after a minute, the addition of 600 µL 20% Na_2_CO_3_. The test tubes were left for 2 h at room temperature and absorbance was read at 765 nm wavelength. The total phenolics were calculated as gallic acid equivalents (GAE) using a standard curve and were presented as mg g^−1^ FW. The standard curve was prepared with gallic acid (Sigma-Aldrich, St. Louis, MO, USA) in the range 0–500 mg L^−1^. 

#### 4.2.6. Determination of Antiradical Activity

Antioxidant activity was measured in the extracts obtained for the total phenolics and 2,2-diphenyl-1-picrylhydrazyl (DPPH) [98]. The incubation mixture contained 100 µL plant extract and 3.9 mL 6 × 10^−5^ mol L^−1^ DPPH (0.06 µmol). The measurement of extract absorption was performed twice: first, immediately after mixing the components (0 min); second, after 30 min at 515 nm and when a parallel blank sample that contained only distilled water was used. The antiradical activity was expressed as % discoloration according to Equation (1): 1 − (A_30min_/A_0min_) × 100.(1)

The DPPH radical scavenging activity of the sample was also expressed as mg Trolox equivalent antioxidant capacity (TEAC) by the formula obtained from the standard curve (y = −0.1954x + 0.708R^2^ = 0.9858). The Trolox was used as a standard.

### 4.3. Anatomical Study of the Leaves

Anatomical observations, for each light treatment, were performed on twenty leaves taken from the 3rd or 4th nodes of the in vitro plants. Small segments (4–5 mm^2^) from the middle part of the leaf lamina were fixed in 3% (*m*/*v*) glutaraldehyde in 0.1 M sodium phosphate buffer (pH 7.4) for 12 h at 4 °C. Several transverse handmade sections mounted on slides with glycerol were used as materials for the histological study. The observations were carried out with Amplival 4 light microscope (Carl Zeiss, Jena, Germany). An average number of fifteen microphotographs with high resolution (2560 × 1960 pixels) were taken with the EcoBlue digital microscope (EC.1657), with an integrated 5.0 MP USB-2 camera (Euromex, Arnhem, The Netherlands) with 400× magnification. The microphotographs were used also for the leaf’s anatomy observation and for the measurements of the lamina (LL), mesophyll (M), palisade parenchyma (PP), spongy parenchyma (SP), and both the adaxial (AdE) and abaxial (AbE) epidermis. Dimensions were presented in µm according to the recommendations of the ImageJ (National Institutes of Health, Bethesda, MD, USA). The stomata observations were performed by using small fragments from the central parts of five leaves, which were preliminary rinsed with distilled water, bleached and rinsed again. The prepared samples were placed on microscopic slides and covered with glycerol. The estimation of the length and width (in µm) of the stomata, as well as the stomatal distribution (number/mm^2^), was based on 30 measurements performed with an Amplival 4 light microscope (Carl Zeiss, Jena, Germany) for each light treatment.

### 4.4. Data Recording and Statistical Analysis

Statistical analyses were performed using one-way analysis of variance (ANOVA), followed by a comparison of group means (Tukey’s b test) with the program SPSS ver. 26.0 (IBM).

## 5. Conclusions

In the present study, LED sources with a different light spectra distinctively affected the growth and development of in vitro cultivated plum plantlets (*Prunus domestica* subsp. *insititia* ‘Saint Julien’). The plantlets exhibited specific growth needs during the different stages of their micropropagation. The results from the current study showed that the plantlets cultivated under the mixed LED light (WBR) at the multiplication stage had the highest fresh (FW) and dry (DW) biomass in comparison to the control plants and to the other light sources. In addition, there was a trend for greater stem length under the mixed light (WBR). Considering these three indicators, we could conclude that under the mixed light, the microplants were of better quality and therefore mixed light (WBR) was more suitable during the multiplication phase. The red LED light stimulated the rooting. The current study protocol can be considered as an effective and affordable method for the large-scale propagation of valuable plum genotypes, which is also suitable for commercial purposes. Furthermore, it can be applied in future research aiming for the successful in vitro cultivation of other ‘difficult-to-propagate’ *Prunus* species or other woody plants.

## Figures and Tables

**Figure 1 plants-12-02125-f001:**
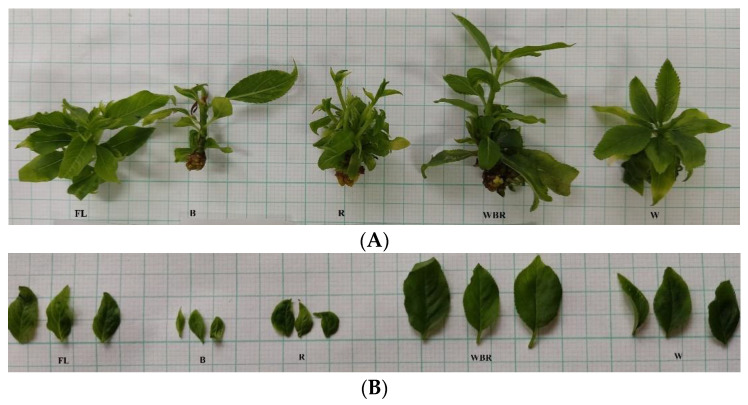
Plum plantlets (**A**) grown under different light treatments and their leaves. (**B**) FL—fluorescence lamps (control), B—blue LED, R—red LED, WRB—mixed LED, W—white LED.

**Figure 2 plants-12-02125-f002:**
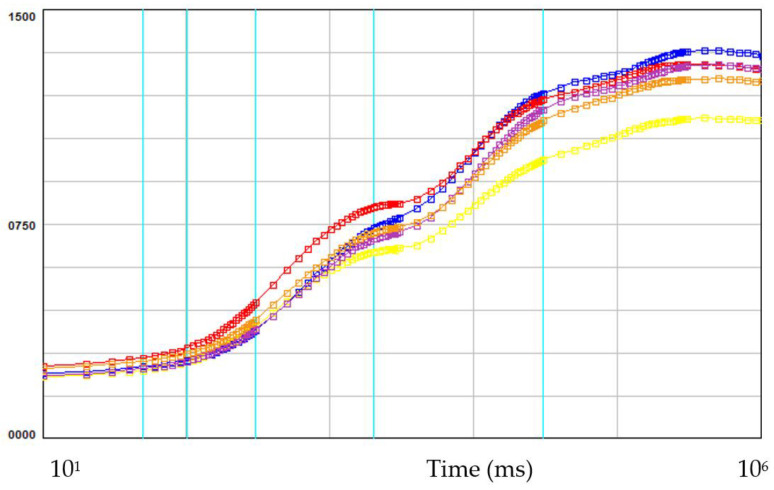
The induction curves of the rapid chlorophyll *a* fluorescence (OJIP test) in plum plantlets grown under different light regimes. **FL**—fluorescence lamps (control), **B**—blue LEDs, **R**—red LEDs, **WRB**—mixed LEDs, **W**—white LEDs.

**Figure 3 plants-12-02125-f003:**
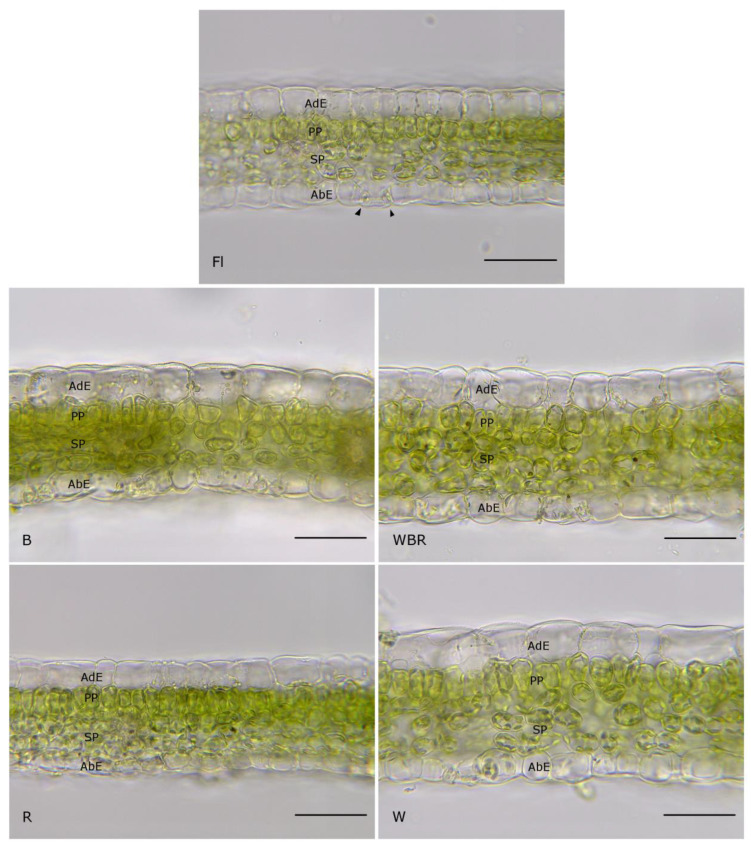
Leaf anatomy of the plum plantlets, cultivated in vitro under different light regimes. Scale bar = 50 µm. Abbreviations used are as follows: AdE—adaxial epidermis, PP—palisade parenchyma, SP—spongy parenchyma, AbE—abaxial epidermis, arrowhead—guard cell; FL—fluorescence lamps (control), B—blue LEDs, R—red LEDs, WRB—mixed LEDs, W—white LEDs.

**Figure 4 plants-12-02125-f004:**
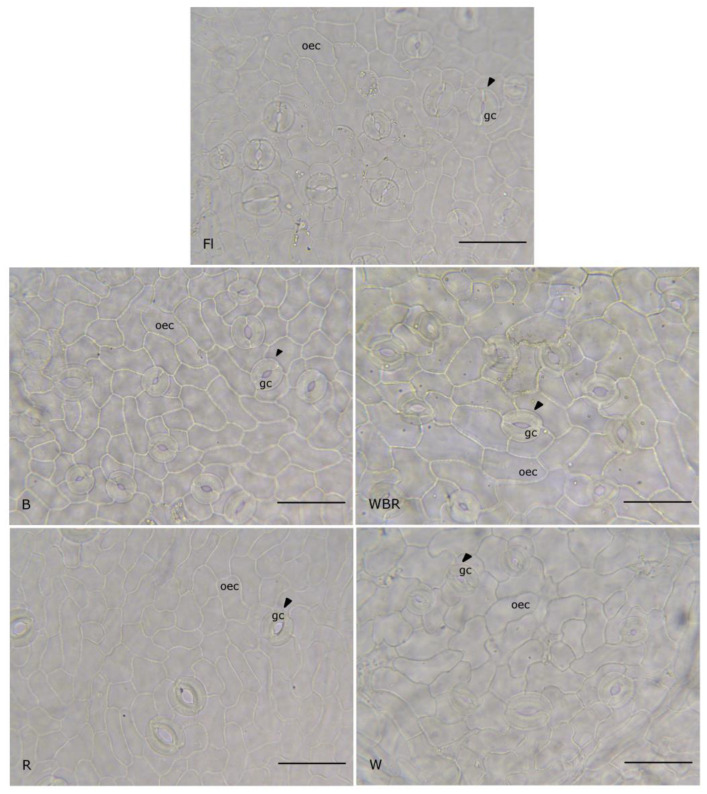
Leaf epidermis and stomata development in the plum plantlets, cultivated under different light regimes. Scale bar = 50 µm. FL—fluorescence lamps (control), B—blue LEDs, R—red LEDs, WRB—mixed LEDs, W—white LEDs, oec—ordinary epidermal cell, gc—guard cell, arrowhead—stoma.

**Figure 5 plants-12-02125-f005:**
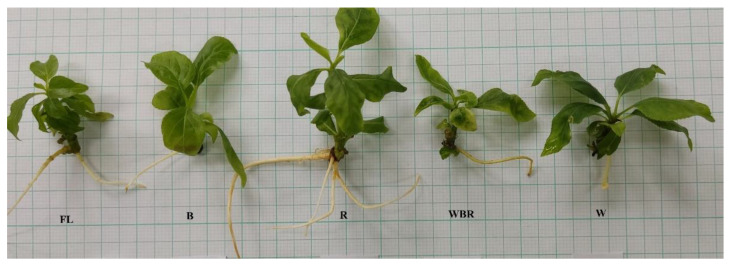
Rooting of in vitro cultivated plum plantlets under different light regimes. FL—fluorescence lamps (control), B—blue LED, R—red LED, WRB—mixed LED, W—white LED.

**Table 1 plants-12-02125-t001:** Growth parameters and multiplication index (MI) of in vitro cultivated plum plantlets under different light sources.

Light Treatment/Parameter	FL	Blue	Red	WBR	White
MI	12.7 b	8.3 c	16.3 a	9.0 c	10.7 b
Stem length (mm)	12.2 bc	10.5 c	16.5 a	15.1 ab	13.1 bc
Number of leaves	25.6 ab	16.7 c	30.6 a	21.4 bc	28.1 ab
Plant FW (mg)	530.4 b	255.3 d	416.0 c	762.9 a	352.0 c
Plant DW (mg)	48.0 b	32.3 c	46.0 b	62.3 a	47.6 b
Leaf length (mm)	18.7 a	8.6 b	8.6 b	16.2 a	14.4 a
Leaf width (mm)	8.8 a	5.0 b	3.8 b	9.7 a	8.1 a

(1) Means followed by the same letter were not different at *p* ≤ 0.05. (2) Abbreviations are as follows: fresh weight (FW) and dry weight (DW). FL—fluorescent lamps (control), WRB—mixed LED, and multiplication index (MI) was calculated as a number of proliferated shoots from one explant.

**Table 2 plants-12-02125-t002:** Photosynthetic pigments content (mg g^−1^ FW) of in vitro cultivated plum plantlets under different light regimes.

Light Treatment/Parameter	FL	Blue	Red	WBR	White
Chl *a*	0.626 ± 0.039 ab	0.815 ± 0.056 a	0.611 ± 0.055 ab	0.568 ± 0.001 b	0.526 ± 0.091 b
Chl *b*	0.186 ± 0.012 a	0.212 ± 0.011 a	0.207 ± 0.105 a	0.138 ± 0.005 a	0.137 ± 0.026 a
Chl (*a* + *b*)	0.811 ± 0.051 ab	1.026 ± 0.68 a	0.817 ± 0.160 ab	0.705 ± 0.005 b	0.662 ± 0.117 b
Car	0.234 ± 0.012 ab	0.306 ± 0.018 a	0.218 ± 0.016 b	0.240 ± 0.015 ab	0.202 ± 0.030 b
Chl (*a*/*b*)	3.362 ± 0.014 a	3.843 ± 0.058 a	3.302 ± 1.407 a	4.100 ± 0.166 a	3.845 ± 0.064 a
Chl/Car	3.466 ± 0.378 a	3.350 ± 0.237 a	3.779 ± 1.010 a	2.947 ± 0.217 a	3.273 ± 0.098 a

(1) Means ± standard error (SE); means followed by the same letter were not different at *p* ≤ 0.05 (*n* = 3). (2) Abbreviations used are as follows: Chl—chlorophyll, Car—carotenoids, FL—fluorescent lamps (Control), WBR—mixed LED.

**Table 3 plants-12-02125-t003:** Net photosynthesis rate (A, µmol CO_2_ plantlet^−1^ s^−1^), transpiration intensity (E, mmol H_2_O plantlet^−1^ s^−1^), photosynthetic water use efficiency (A/E, µmol CO_2_ mmol^−1^ H_2_O) and stomatal conductance (gs, plantlet^−1^ s^−1^) of in vitro cultivated plum plantlets under different light regimes.

Light/Parameter	A	E	A/E	gs
FL	4.04 ± 0.18 ab	1.12 ± 0.18 c	3.61 ± 0.41 a	0.078 ± 0.007 c
B	2.93 ± 0.21 c	1.14 ± 0.07 c	2.27 ± 0.03 b	0.055 ± 0.007 d
R	3.97 ± 0.61 ab	1.63 ± 0.13 a	2.44 ± 0.18 b	0.097 ± 0.014 ab
WBR	3.78 ± 0.08 b	1.54 ± 0.06 b	2.45 ± 0.05 b	0.085 ± 0.005 b
W	4.43 ± 0.41 a	1.65 ± 0.01 a	2.68 ± 0.24 b	0.113 ± 0.014 a

(1) Means ± standard error (SE); means followed by the same letter were not different at *p* ≤ 0.05 (*n* = 5). (2) Abbreviations used are as follows: FL—fluorescence lamps (control), B—blue LEDs, R—red LEDs, WBR—mixed LEDs, W—white LEDs.

**Table 4 plants-12-02125-t004:** Content of some antioxidant substances of in vitro cultivated plum plantlets under different light regimes.

Light Treatment	% DPPH Inhibition	mg TE/100 mL	GAE mg/g FW
FL	21.18 e	4.54 d	0.58 c
B	26.12 b	7.52 c	1.65 b
R	23.05 d	7.95 b	1.74 b
WBR	44.09 a	8.39 a	4.78 a
W	24.83 c	7.77 c	1.71 b

(1) Means followed by the same letter were not different at *p* ≤ 0.05 (*n* = 3). (2) Abbreviations used are as follows: total polyphenols (mg GAE/g FW); antiradical activity (mg TE/100 mL); antiradical activity (% DPPH/g FW); FL—fluorescence lamps (control), W—white LEDs, R—red LEDs, B—blue LEDs, WBR—mixed LEDs.

**Table 5 plants-12-02125-t005:** Parameters of the rapid chlorophyll fluorescence (OJIP test) of the in vitro cultivated plum plantlets under different light regimes.

Parameter/Light Treatment	FL	B	R	WBR	W
F_0_	247 ± 22 a	233 ± 22 a	249 ± 48 a	221 ± 34 a	223 ± 39 a
F_M_	1263 ± 55 a	1363 ± 114 a	1312 ± 121 a	1310 ± 199 a	1187 ± 193 a
F_V_	1017 ± 35 a	1130 ± 143 a	1063 ± 75 a	1090 ± 193 a	964 ± 154 a
F_V_/F_M_	0.805 ± 0.009 c	0.828 ± 0.006 ab	0.811 ± 0.019 bc	0.831 ± 0.005 a	0.813 ± 0.004 bc
ψE_0_	0.53 ± 0.05 ab	0.55 ± 0.04 a	0.47 ± 0.02 b	0.55 ± 0.06 a	0.53 ± 0.01 ab
φE_0_	0.43 ± 0.03 ab	0.46 ± 0.03 a	0.38 ± 0.02 b	0.46 ± 0.05 a	0.43 ± 0.01 ab
δR_0_	0.28 ±0.05 ab	0.24 ± 0.03 b	0.25 ± 0.05 b	0.25 ± 0.05 b	0.34 ± 0.01 a
PI_abs_	3.23 ± 0.83 b	5.04 ± 0.91 a	2.31 ± 0.53 b	5.06 ±0.98 a	2.93 ± 0.31 b
PI_total_	1.20 ± 0.10 b	1.63 ± 0.16 a	0.74 ± 0.14 c	1.85 ± 0.31 a	1.50 ± 0.17 a

Means ± standard error (SE); means followed by the same letter were not different at *p* ≤ 0.05 (*n* = 5). Abbreviation used are as follows: FL—fluorescence lamps (control), B—blue LEDs, R—red LEDs, WBR—mixed LEDs, W—white LEDs.

**Table 6 plants-12-02125-t006:** Leaf morphometric parameters of the in vitro cultivated plum plantlets under different light regimes.

Thickness (µm)	FL	B	R	WBR	W
Leaf lamina	78.03 a	79.90 a	73.36 b	90.96 c	91.19 c
Mesophyll	43.49 a	42.26 a	44.06 a	50.40 b	52.43 b
Palisade parenchyma	15.26 a	14.68 a	14.86 a	15.51 a	18.97 b
Spongy parenchyma	29.73 a	27.14 b	28.42 ab	34.33 ac	35.55 c
Adaxial epidermis	20.00 a	22.57 b	17.35 c	24.33 b	22.52 b
Abaxial epidermis	14.80 a	14.65 a	13.14 b	16.93 c	15.72 a

Means followed by the same letter were not different at *p* ≤ 0.05 (*n* = 30). Abbreviations used are as follows: FL—fluorescence lamps (control), B—blue LEDs, R—red LEDs, WRB—mixed LEDs, W—white LEDs.

**Table 7 plants-12-02125-t007:** Effect of different light sources and LED wavelength on the stomata frequency (number per mm^2^) and stomata size (μm) of in vitro cultivated plum plantlets.

Light/Parameter	FL	B	R	WBR	W
Stomata length (μm)	24.2 a	25 a	24.8 a	29.1 b	26.5 ab
Stomata width (μm)	21.9 a	21 a	23.1 ab	25.4 b	23.2 ab
Stomata frequency (per mm^2^)	182 a	139 b	89 c	163 ab	164 ab

Means followed by the same letter were not different at *p* ≤ 0.05 (*n* = 30). Abbreviations used are as follows: FL—fluorescence lamps (control), B—blue LEDs, R—red LEDs, WRB—mixed LEDs, W—white LEDs.

**Table 8 plants-12-02125-t008:** Rooting, numbers of roots and mean root length of in vitro cultivated plum plantlets under different light regimes.

Light/Parameter	Rooting (%)	Mean Number of Roots	Mean Root Length (mm)
FL	68.42	1.53 ± 0.61 a	19.86 ± 4.09 c
B	62.7	2.06 ± 0.64 a	20.53 ± 3.96 c
R	98.67	2.03 ± 0.82 a	32.85 ± 2.11 a
WBR	19.04	1.40 ± 0.48 a	27.71 ± 2.58 b
W	65.6	1.70 ± 0.55 a	21.23 ± 4.12 c

Means ± standard error (SE); means followed by the same letter were not different at *p* ≤ 0.05 (*n* = 40). Abbreviations used are as follows: FL—fluorescence lamps (control), B—blue LEDs, R—red LEDs, WRB—mixed LEDs, W—white LEDs.

**Table 9 plants-12-02125-t009:** Definitions of the recorded and calculated chlorophyll *a* fluorescence parameters according to Strasser and Strasser [93] and Goltsev [94].

Parameter	Description
F_0_~F20 µs	Minimum fluorescence, when all PSII reaction centers (RCs) are open
F_J_	Fluorescence at the J-step (2 ms) of the O-J-I-P transient
F_I_	Fluorescence at the I-step (30 ms) of the O-J-I-P transient
F_M_ = F_P_	Maximum recorded fluorescence at the P-step when all RCs are closed
V_J_ = (F_J_ − F_0_)/(F_M_ − F_0_)	Relative variable fluorescence at the J-step
F_V_ = F_M_ − F_0_	Variable fluorescence
ψ_E0_ = 1 − V_J_	Probability (at t = 0) that a trapped exciton moves an electron into the electron transport chain beyond QA^−^
φ_E0_ = (1 − F_J_/F_M_)	Quantum yield (at t = 0) for electron transport from QA^−^ to plastoquinone
δR_0_ = (1 − V_I_)/(1 − V_J_)	Efficiency/probability (at t = 0) with which an electron from the intersystem carriers moves to reduce end electron acceptors at the PSI acceptor side
PI_ABS_	Performance index of PSII based on absorption
PI_total_ = PI_ABS_ × δR_O_/(1 − δR_O_)	Performance index of electron flux to the final PSI electron acceptors, i.e., of both PSII and PSI

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
