# Peer review of "In Vitro Multiplication and Rooting of Plum Rootstock ‘Saint Julien’ (Prunus domestica subsp. insititia) under Fluorescent Light and Different LED Spectra"

_plants, 2023, doi:10.3390/plants12112125_

Round 1
Reviewer 1 Report
The results here presented, on the effects of different light systems on either morphological, physiological, and biochemical aspects of in vitro culture of plum, are highly valuable. This’s an important and controversial topic, not only for the different effects reported on in vitro cultured plants, but also is of great concern for commercial micropropagation because its economic implications. Undoubtedly this research will contribute to shed light on a more efficient use of light systems in in vitro culture labs. However, there’re some aspects that can be improved, aimed to increase the comprehension of research, and their extents.
Since multiplication index is highly important for commercial micropropagation, as was set in text, the effects of different light regimes on it must be presented in Abstract.
What’s multiplication index, and how it was determined?
Why PPFD was set in 87 ± 7.5 μmol m–2 s–1?
It’s important to set how fresh and dry weight were determined, and which part of “microplants” were used for. It’s important to shed light to the apparent contradiction that the mixed lights produced the highest biomass; however, had the lowest multiplication index (along with blue light).
Some parts of results should be moved to Discussion, e g, from line 161 to 167, 202 to 217, and so on. In this section focus the attention only on results. In table 4 it would be recommendable to include the WUE calculations. Since water lost has been signaled as the main cause of dead during acclimation, has light quality some influences on survival under ex vitro conditions? It would be worth to show results of acclimation, if it’s possible. In figure 3, show which's both the abaxial and adaxial side. Also indicate the guard cells.
For Discussion section, I would suggest only to discuss the obtained results. The first part (from line 345 to 372) is more proper for Introduction. Again, I missed a depper discussion on repercussion of the different light systems used on the results of acclimation.
Author Response
Dear Reviewer,
We thank for your suggestions that will improve our manuscript.
We accept them and have reflected them in the revised text.

Reviewer 2 Report
The manuscript can be of interest to wide readers of journals and contributes to existing knowledge on the subject matter. However, I have pointed out few pertinent points for improving the clarity of the content and boosting the scientific soundness of the manuscript.
Minor comments
Title (line 2) and line 39
Replace “Prunus domestica ssp. Insititia” by “Prunus domestica ssp. insititia”
Line 15: scientific name should be in italic form. Maintained it throughout the manuscript.
“Prunus domestica”
Line 19: Format of unit is wrong
Replace “μmol m–2 s–1” by “μmol m–2 s–1”
Abstract is incomplete. Authors need to present findings as integral values or percentage advantage
Introduction
More information may be added on pertinence of Prunus domestica ssp. insititia interms of its food values, medicinal values, traditional uses, industrial uses etc.
Results
Table 1 and 2. Revise the footnotes.
There is no B, R in the table.
Check “B – blue LED; R – red LED”
Abbreviations are as follow: Fresh weight (FW) and dry weight (DW). FL - Fluorescent lamps 96 (Control), B – blue LED; R – red LED; WRB – mixed LED; W white LED.
Table 2. Overlapped footnotes and table. Correct it.
Table 1 and 2: Keep to digits after decimal. Maintain uniformity.
Line 206 and 209: Avoid using references in the “results sections”
Adjust citations such as 35-45, in the discussions section
Figure 2 is not clear. Too compressed. Submit more distinct pictures and data.
Figure 3 and 4: Leaf anatomy of the plum plantlets. Label all the different cells and tissues observed in the figures and discuss it in the manuscript.
Discussion
Line 345 -371: Author already mentioned the advantages of using LEDs over conventional light sources in invitro culture. Avoid repeating it in the discussion sections.
Line 437-439: Missing citations. Keep citations for the statements.
Line 499: Donot keep unpublished data and statements.
Line 500-501: Missing citations. Keep citations for the statements.
Major comments
1. Mechanism involved in the changes in the physiological, anatomical and chemical properties under different light (LEDs) sources is missing in the DISCUSSION section.
2. Phytochemical analysis using HPLC, LC-MS/MS etc can be used to find the difference in the
Phytochemical profile in the plants under different light sources.
Author Response

(The authors gave the same response as above.)

Reviewer 3 Report
LEDs have become a valid alternative in the plant tissue culture; the authors have reported the studies performed in this work and have well structured the discussion of results that are clearly presented. The authors have cited references which are all fairly recent.
Author Response
Dear Reviewer
Thank you very much for your high evaluation of our work and publication.
Round 2
Reviewer 2 Report
Comments
The author has revised the manuscript satisfactorily. There are some minor issues in the manuscript.
Check table 2. Its not done properly. Keep demarcation (line) below the different types of light source.
Table 6: overlapping table. Keep it in one page, not divided into two page
Materials and methods:
Check the format of journal. Subheadings should be numbered.
Author Response
Dear Reviewer,
We thank for your suggestions that will improve our manuscript.
We are very sorry for the technical inaccuracies.
We have fixed them in the revised text.